# Bone Density, Geometry, Structure and Strength Estimates in Adolescent and Young Adult Women with Atypical Anorexia Nervosa versus Typical Anorexia Nervosa and Normal-Weight Healthy Controls

**DOI:** 10.3390/nu15183946

**Published:** 2023-09-12

**Authors:** Shubhangi Tuli, Vibha Singhal, Meghan Slattery, Nupur Gupta, Kathryn S. Brigham, Jennifer Rosenblum, Seda Ebrahimi, Kamryn T. Eddy, Karen K. Miller, Madhusmita Misra

**Affiliations:** 1Neuroendocrine Unit, Massachusetts General Hospital and Harvard Medical School, 55 Fruit Street, Boston, MA 02114, USA; 2Division of Pediatric Endocrinology, Massachusetts General Hospital and Harvard Medical School, 55 Fruit Street, Boston, MA 02114, USA; 3MGH Weight Center, Massachusetts General Hospital, 55 Fruit Street, Boston, MA 02114, USA; 4Division of Adolescent Medicine, Massachusetts General Hospital and Harvard Medical School, Boston, MA 02114, USA; 5Cambridge Eating Disorders Center, Cambridge, MA 02138, USA; 6Eating Disorders Clinical and Research Program, Massachusetts General Hospital, Boston, MA 02114, USA; keddy@mgh.harvard.edu

**Keywords:** anorexia nervosa, atypical anorexia nervosa, adolescents, bone mineral density, bone microarchitecture, bone geometry, bone strength

## Abstract

Our objective was to characterize bone outcomes in adolescent and young adult women with atypical anorexia nervosa (AAN) compared to typical AN and normal-weight healthy controls (HC) based on DSM-5 criteria. Four hundred thirty-two participants (141 AN, 131 AAN and 160 HC), ages 12–21 years, underwent dual-energy X-ray absorptiometry for areal BMD, and a subset had high-resolution peripheral quantitative CT assessment of the distal radius and tibia for volumetric BMD (vBMD), bone geometry and microarchitecture, and microfinite element analysis for estimated strength. The groups did not differ for age, pubertal stage, menarcheal age or physical activity. BMI and bone outcomes overall were intermediate in AAN compared with AN and HC. This applied to spine, total hip and femoral neck BMD measures and many distal tibial measures. However, the mean whole-body less head BMD Z-score did not differ between AAN and AN, and it was lower in both vs. HC. Similarly, many distal radius measures did not differ between AAN vs. AN or HC but were lower in AN than HC. Lower BMI, lean mass and bone age, older menarcheal age and longer illness duration correlated with greater impairment of bone outcomes. These data indicate that individuals with AAN overall have bone outcomes that are intermediate between AN and HC.

## 1. Introduction

While the deleterious effect of anorexia nervosa (AN) on bone health is now well established, the impact of atypical anorexia nervosa (AAN) on bone health has not been as well characterized. In 2013, the Diagnostic and Statistical Manual of Mental Disorders-Fifth Edition (DSM-5) was published and included a revised definition for AN and also included AAN as a specific example of an ‘Other Specified Feeding and Eating Disorder’ (OSFED) [1]. The new criteria for AN no longer require the presence of amenorrhea and do not provide a specific weight threshold for the diagnosis of AN. The category AAN includes patients who are not low weight but meet other criteria for AN and have sustained weight loss over time. Importantly, data demonstrate that symptom severity of AAN is as severe as, and sometimes more severe than, AN [2].

Multiple studies have been published over the years describing low areal bone mineral density (aBMD) using dual-energy X-ray absorptiometry (DXA), low volumetric BMD (vBMD) and altered bone geometry using peripheral quantitative computed tomography (pQCT) and high-resolution pQCT (HRpQCT), impaired bone microarchitecture using HRpQCT, and low strength estimates using microfinite element analysis (µFEA) in adults and adolescents with AN compared with normal-weight healthy controls using the older DSM-IVR criteria [3,4,5,6,7,8,9]. However, only a handful of studies have examined the impact of AN, as defined in DSM-5, and particularly of AAN, on bone outcomes [10,11,12,13]. This is particularly important to establish in adolescents, who are in the process of accruing bone mass towards attainment of optimal peak bone mass, a key determinant of bone health and fracture risk in future life [14,15].

Nagata et al. examined differences in aBMD using DXA in adolescent AN (n = 263) vs. AAN (n = 23) and reported higher aBMD Z-scores in AAN at the lumbar spine and femoral neck and higher whole-body bone mineral content Z-scores, fat mass index and lean mass index in those with AAN [12]. They also demonstrated that higher prior BMI Z-scores were associated with higher BMD Z-scores at all sites. However, this study did not compare girls with AAN to normal-weight healthy controls (HC), which is important in order to ascertain whether normal-weight in AAN is protective against bone loss. Further, this study did not assess vBMD, bone geometry, structure or strength estimates, all of which are important determinants of fracture risk. Pehlivanturk-Kizilkan et al. examined predictors of BMD in AAN alone and similarly reported positive associations of lifetime maximum BMI with lumbar spine and femoral neck BMD, and positive associations of BMI at admission and inverse associations of duration of amenorrhea with lumbar spine BMD Z-scores, similar to previous reports in AN [10]. However, this study was also limited to DXA measures of aBMD and did not compare the AAN group to either AN or HC. 

In order to address existing knowledge gaps, we compared bone outcomes using DXA, HRpQCT and µFEA in adolescent and young adult women 12–21 years old meeting DSM-5 criteria for AAN and AN as well as HC. Based on known associations of BMD Z-scores with BMI, we hypothesized that bone outcomes would be less severely impacted in those with AAN than AN; however, adolescents and young adults with AAN would demonstrate more deleterious outcomes than normal-weight controls. We also hypothesized that the proportion of participants with low BMD Z-scores would be highest in AN, intermediate in AAN and lowest in HC.

## 2. Materials and Methods

This cross-sectional study included 432 participants 12–21 years old, 141 with AN, 131 with AAN and 160 HC drawn from participants enrolled in earlier studies that included data from DXA and/or HRpQCT and µFEA scans between 2000 and 2018 [5,16,17,18]. These studies screened a large number of adolescents and young adult women with restrictive eating disorders across a range of body mass index (BMI) and varying menstrual status. Because many participants had initially been categorized per DSM-IV criteria, all participants were recategorized per DSM-5 criteria for this analysis, taking into account their body mass index (BMI) and menstrual history. Specifically, the new criteria for AN no longer require the presence of amenorrhea and do not provide a specific weight threshold for the diagnosis of AN. Thus, participants were grouped as having AN if they had a BMI of <18.5 kg/m^2^ if ≥18 years old, or <5th percentile if <18 years old, along with other features of AN regardless of menstrual status. Participants were grouped as having AAN if they had a BMI of ≥18.5 kg/m^2^ if ≥18 years old, or ≥5th percentile if <18 years old, along with other features of AN (and including a history of weight loss) but regardless of menstrual status. Healthy controls were between the 10th and 90th percentiles for age for BMI, eumenorrheic when postmenarchal, and had no lifetime history of eating disorders. Participants were recruited for the respective studies through mailings to pediatricians, adolescent medicine providers, psychologists, psychiatrists, dieticians, and eating disorder programs, through advertisements in local newspapers, and flyers placed around our institution and in area colleges. The exclusion criteria for the study included conditions other than eating disorders that might impact bone health and use of medications (other than calcium and vitamin D supplements) within 3 months of the study visit that might impact bone outcomes. Per DSM-5, the study sample includes participants who were normally menstruating, had irregular periods, or had amenorrhea.

All protocols from which participants were drawn for this study were approved by our Institutional Review Board and were Health Insurance Portability and Accountability Act compliant. Participants 18 years or older provided informed consent for the study; parents of those under 18 years old provided informed consent while the participants provided their assent for study participation.

All participants were measured for height using a wall-mounted stadiometer in triplicate and the mean recorded; weight was measured on an electronic scale to the nearest 0.1 kg. Body mass index (BMI) was calculated as weight in kg/(height in meters)^2^. Percent median BMI was based on the participant’s BMI in relation to the 50th percentile of BMI for age from CDC growth data for females [19]. Activity level in hours per week was assessed using the Modifiable Activity Questionnaire [20].

We used DXA (Hologic 4500A, Waltham, MA, USA) to assess aBMD (lumbar spine, total hip, femoral neck, whole-body and whole-body less head (WBLH)) and body composition. All BMD assessments were performed at our Translational and Clinical Research Center by our Nutrition and Metabolism Core by ISCD-certified personnel. Not all assessments are available for all participants. Some studies did not include the hip or whole-body assessment, and some scans were not usable. The numbers of participants who had BMD assessed at any specific site are indicated in the respective table. The coefficients of variation for areal BMD, fat mass and lean mass measurements are 0.8% to 1.1%, 2.1% and 1.0%, respectively, for our institution. The Longitudinal Childhood Bone Mineral Density database was used to calculate BMD Z-scores and height Z-score (HAZ) adjusted BMD Z-scores for participants 20 years and younger [21]. BMD Z-scores of ≤−2 are considered to be low for all ages, and BMD Z-scores of <−1 are considered to be low for those engaged in weight-bearing exercise [22,23].

HRpQCT (XtremeCT, Scanco Medical AG, Brüttisellen, Switzerland) was used to assess vBMD, bone geometry and structure at the distal radius and tibia using previously published methods [3,6] and was available for a subset of 138 participants (23 AN, 37 AAN and 78 HC). Briefly, scans were performed on the non-dominant side unless there was a history of a fracture on that side, in which case the non-fractured dominant side was scanned. Slices were assessed 9.5 mm from the distal radial endplate and 22.5 mm from the distal tibial endplate with the help of 2D scout films obtained as part of the protocol. We report measures of cortical and trabecular area, cortical thickness and porosity, trabecular thickness, number and separation, and cortical, trabecular and total vBMD at the distal radius (site of non-weight-bearing bone) and distal tibia (site of weight-bearing bone). µFEA was used to assess strength estimates (stiffness and failure load) and elastic modulus using published methods [3,24]. The load-to-strength ratio (factor of risk) at the distal radius was calculated to assess the risk of fracture at the distal forearm by relating applied force of fall to bone strength [25]. A higher load-to-strength ratio is consistent with greater risk of fracture. Force of fall was calculated using the formula (height in meters × weight in kg × 9.81 × 4527 × 2)^0.5^ [26]. The estimated failure load from µFEA was used as the measure of bone strength. This is the simulated axial compression force that results in >2% of bone tissue experiencing >7000 µstrain [27].

Data analysis was performed using JMP statistical software (version 16). All data are reported as mean ± standard error of the mean (SEM) or median (first quartile, third quartile) depending on data distribution. Continuous data were compared across groups using analysis of variance (ANOVA), followed by the Tukey Kramer test to adjust for multiple comparisons across groups when data were normally distributed, or the Kruskal–Wallis test followed by the Steel–Dwass test to adjust for multiple comparisons across groups when data were not normally distributed. We adjusted for race when comparing HRpQCT and µFEA measures using multivariate regression (this was not required for DXA measures as calculation of BMD Z-scores takes race into account). Categorical variables were compared using the Fisher’s exact test. Associations between bone variables and possible determinants of these variables were assessed using Pearson’s correlation. We set a *p*-value of <0.05 a priori to indicate significance.

## 3. Results

### 3.1. Clinical Characteristics

The groups did not differ for age, bone age, height, Tanner stage, menarcheal age, or exercise activity (Table 1). AN and AAN groups did not differ for duration of amenorrhea or duration since diagnosis. As expected and per study design, weight, BMI and percent median BMI were lower in AN and AAN than HC, and in AN than AAN (Table 1). Lean mass, fat mass and percent body fat, similarly, were lower in AN and AAN than HC, and in AN than AAN (Table 1). The majority of participants were White (87.4%, 92.0% and 70.3% of AN, AAN and HC, respectively) or Asian (3.9%, 0% and 13.3% of AN, AAN and HC, respectively). Racial distribution differed significantly across groups; hence, we controlled for race in our analysis of HRpQCT and µFEA variables (DXA Z-scores take race into account).

### 3.2. Areal Bone Mineral Density Measures (as Assessed by DXA)

Spine BMD was lower in AN and AAN than HC but did not differ between AN and AAN. However, spine height-adjusted BMD Z-scores were lower in AN and AAN than HC (Table 1 and Figure 1). Spine height-adjusted BMD Z-scores were ≤−2 in 18.1%, 10.8% and 9.0% in AN, AAN and HC, respectively, and between −1 and −2 in 37.1%, 36.9% and 18.8% of those with AN, AAN and HC, respectively (*p* < 0.001) (Figure 2).

Total hip BMD and height-adjusted BMD Z-scores were lower in AN and AAN than HC, and in AN vs. AAN (Table 1 and Figure 1). Total hip height-adjusted BMD Z-scores were ≤−2 in 16.7%, 9.5% and 2.0% in AN, AAN and HC, respectively, and between −1 and −2 in 25.5%, 24.8% and 16.7% in AN, AAN and HC, respectively (*p* < 0.001) (Figure 2). Femoral neck BMD and height-adjusted BMD Z-scores were lower in AN vs. HC, and femoral neck BMD was lower in AAN than HC (Table 1 and Figure 1). Femoral neck height-adjusted BMD Z-scores were ≤−2 in 17.3%, 8.8% and 3.9% in AN, AAN and HC, respectively, and between −1 and −2 in 33.3%, 28.5% and 18.6% in AN, AAN and HC, respectively (*p* < 0.001) (Figure 2).

Whole-body BMD and height-adjusted BMD Z-scores were lower in AN and AAN than HC but did not differ between AN and AAN (Table 1 and Figure 1). Whole-body height-adjusted BMD Z-scores were ≤−2 in 13.2%, 14.0% and 6.8% in AN, AAN and HC, respectively, and between −1 and −2 in 39.7%, 28.1% and 24.1% in AN, AAN and HC, respectively (*p* = 0.006) (Figure 2). Whole-body less head BMD and height-adjusted BMD Z-scores were lower in AN than HC but did not differ between AN and AAN (Table 1 and Figure 1). Whole-body less head BMD was lower in AAN than HC. Whole-body less head height-adjusted BMD Z-scores were ≤−2 in 20.7%, 17.0% and 9.8% in AN, AAN and HC, respectively, and between −1 and −2 in 36.6%, 27.7% and 29.4% in AN, AAN and HC, respectively (*p* = 0.089) (Figure 2).

Within the AN and AAN groups, participants who presented with amenorrhea did not differ from those who did not present with amenorrhea for areal BMD height-adjusted Z-scores.

### 3.3. HRpQCT and Microfinite Element Analysis Measures

Distal radius: AN had a lower radial cortical area, cortical thickness, trabecular thickness, trabecular and total vBMD, and higher cortical porosity compared to HC but did not differ from AAN; AAN did not differ from HC for these measures (Table 2). Radial stiffness was lower in AN and AAN vs. HC, and failure load was lower in AN vs. AAN and HC (Figure 3). Elastic modulus was lower in AN vs. HC. Differences between AN and HC groups persisted after adjusting for height. After controlling for race, differences between AN and HC persisted for the radial cortical area, trabecular thickness, trabecular and total vBMD, stiffness, failure load and elastic modulus. The factor of risk did not differ significantly across groups.

Distal tibia: Both AN and AAN vs. HC had a lower tibial cortical area, cortical thickness and trabecular number, greater trabecular separation and cortical porosity, and lower tibial stiffness and failure load (Table 2 and Figure 3). All differences, other than for cortical porosity, persisted after controlling for race. Further, AN vs. HC had lower percent cortical area, greater percent trabecular area, and lower cortical, trabecular and total vBMD and elastic modulus. All differences, other than for tibial cortical vBMD, persisted after adjusting for race. Additionally, AN vs. AAN had lower tibial total vBMD, and this difference persisted after adjusting for race. Most differences persisted after adjusting for height.

The numbers of participants were not sufficient to determine whether those who presented with amenorrhea differed from those who did not present with amenorrhea for HRpQCT and µFEA endpoints within the AN and AAN groups.

### 3.4. Associations between Bone Variables and Possible Determinants of These Variables

Table 3 shows associations of bone variables with BMI, percent median BMI, lean mass, bone age, menarcheal age, duration of illness, duration of amenorrhea and exercise activity.

DXA measures of areal BMD: For the group as a whole, BMI and percent median BMI had the strongest positive associations with BMD Z-scores at all sites, followed by lean mass, which was associated significantly with BMD Z-scores at all sites except the spine. In addition, menarcheal age was associated inversely with all measures of BMD Z-scores. In this cohort, we did not find significant associations of BMD Z-scores for the most part with bone age, duration since diagnosis, duration of amenorrhea and exercise activity, except for weak positive associations of bone age with whole-body less head BMD Z-score, and inverse associations of duration of amenorrhea with total hip BMD Z-score.


HRpQCT Measures


Distal radius: At this non-weight-bearing site, BMI, and to a lesser extent percent median BMI, were associated positively with bone geometry, structure and vBMD. Lean mass was associated positively with trabecular thickness and vBMD and failure load, while bone age was positively associated with percent cortical area and thickness, and cortical and total vBMD. Menarcheal age was associated inversely with percent radial cortical area, trabecular thickness, and cortical, trabecular and total vBMD. Duration since diagnosis was inversely associated with trabecular thickness and vBMD, and with failure load. There were no associations of duration of amenorrhea or hours of exercise activity with radial measures.

Distal tibia: At this weight-bearing site, BMI and percent median BMI were associated positively with all measures of geometry, structure (except trabecular thickness) and vBMD, and with failure load. Lean mass was positively associated with cortical thickness, trabecular number and trabecular and total vBMD, and failure load. Bone age also demonstrated positive associations with percent cortical area and thickness, and cortical and total vBMD. Menarcheal age was inversely associated with most measures of geometry, structure (other than trabecular number) and vBMD but not failure load. Further, duration since diagnosis was inversely associated with percent cortical area and thickness, total vBMD and failure load. There were no associations with duration of amenorrhea and hours of exercise activity.

## 4. Discussion

As hypothesized, we found that, overall, AAN had bone outcomes that were better than AN but worse than HC, thus intermediate between AN and HC. This was true for total hip and femoral neck BMD DXA measures and for height-adjusted BMD Z-scores at the spine, total hip and femoral neck. However, this was not consistent across measures. For example, for spine, whole-body and whole-body less head BMD, and for whole-body and whole-body less head height-adjusted BMD Z-scores, measures did not differ between AN and AAN but were lower in both AN and AAN compared to HC. For tibial parameters, AAN vs. HC had a lower tibial cortical area, cortical thickness and trabecular number, greater trabecular separation and cortical porosity, and lower tibial stiffness and failure load (but did not differ from AN for these measures), while AAN had higher tibial total vBMD than AN. For radial parameters, failure load was lower in AN vs. AAN, and radial stiffness was lower in AAN than HC. Similarly, many distal radius measures (by HRpQCT) did not differ between AN and AAN (radial cortical area, thickness and porosity, trabecular thickness, trabecular and total vBMD, elastic modulus) but were adversely impacted in AN vs. HC. Other measures did not differ between AAN and HC, again suggesting that AAN were mostly intermediate between AN and HC for these measures.

One other study by Nagata et al. that included 263 AN and 23 AAN adolescent and young adult participants has reported on DXA measures of areal BMD across these groups but did not compare these groups with HC [12]. This study reported that AAN had higher Z-scores for spine and femoral neck BMD and whole-body bone mineral content than AN. Our data for comparisons of 131 AN and 137 AAN (thus, a much larger number of AAN) are similar in that we too found that AN and AAN differed for spine, total hip and femoral neck BMD and height-adjusted BMD Z-scores. However, we also found that AAN had lower measures for BMD at these sites than 160 HC. Further, in contrast to the study by Nagata et al., we found that whole-body and whole-body less head BMD and height-adjusted BMD Z-scores did not differ between AN and AAN but were lower in both than in HC. Of note, there are some differences in the clinical characteristics of participants with AAN in the study by Nagata et al. compared with our participants, e.g., mean percent median BMI at presentation of 99.7% vs. 90.8% in our study. Thus, the AAN participants in the study by Nagata et al. had a mean percent median BMI that was closer to that of the healthy controls in our study (102.8%). In contrast, our AAN participants were intermediate between AN (78.8%) and healthy controls (102.8%) for mean percent median BMI. This may explain why BMD Z-scores of AAN participants in that study were overall more robust than in AAN in our study. Finally, Nagata et al. did not report measures of vBMD, bone geometry, structure or strength, which we report in this study in a subset of participants. These endpoints are known to be important determinants of fracture risk, and thus add to the literature for adolescents with AAN.

Haines et al. have reported on differences in adult women with AAN and controls, but do not compare these groups with AN [28], and found lower total hip, femoral neck and total radius BMD Z-scores in AAN vs. controls (mean BMI 19.5 kg/m^2^ and 22.2 kg/m^2^, respectively), while spine BMD Z-scores did not differ across groups (the numbers of AAN women (n = 28) and controls (n = 27) were lower in this study than in ours, which might have limited the ability to determine differences across groups for all sites). In the study by Haines et al., 27%, 4% and 11% of AAN participants had BMD Z-scores of <−2 at the spine, total hip and femoral neck, respectively. In our study, 10.2%, 10.9% and 8.3% of AAN participants had BMD Z-scores of <−2 at the respective sites. The lower prevalence of BMD Z-scores of <−2 at the spine in our study may be attributable to the younger age and shorter duration of AAN in our study participants (mean of 9.5 months in our study vs. 11.2 years in the study by Haines et al.). However, it is concerning that despite the much shorter duration of illness in our study participants, so many of our participants had low BMD Z-scores at the various sites. The study by Haines et al. found significantly lower tibial total, cortical and trabecular vBMD, cortical thickness and failure load, and lower radial trabecular number in AAN compared with controls. Similarly, we found lower tibial cortical thickness (but not total, cortical or trabecular vBMD), and also lower trabecular number and higher cortical porosity than controls. In contrast, we found no differences between AAN and control groups for HRpQCT outcomes at the radius, likely also attributable to the shorter duration of disease in our younger population. Our study did find lower measures of estimated bone strength at both sites in those with AAN vs. controls and higher measures of radial failure load in AAN than AN. However, the load-to-strength ratio at the radius did not differ between AAN and AN or controls, indicating that fracture risk at the radius subsequent to a fall is similar in AAN to the other two groups (despite higher failure load), possibly because of the relatively low weight (and thus force of fall) in relation to estimated strength.

Schorr et al. have reported on bone endpoints in males with AN and AAN [11], and reported that AAN in their study did not differ from controls for BMD Z-scores at all sites, even though mean percent ideal body weight was only 86.6% in AAN vs. 99.2% in controls. However, this study included only 18 participants with AAN, which may have limited their ability to detect significant differences between groups; or men with AAN may behave differently from women with AN. This study also reported that 77%, 50% and 58% of AN participants and 43%, 33% and 33% of AAN participants had BMD Z-scores <−1 at the spine, total hip and femoral neck, respectively. In our study, 55.1%, 43,4% and 53.2% of AN participants and 47.5%, 33.6% and 36.1% of AAN participants had BMD Z-scores of −1 at the spine, total hip and femoral neck, respectively. Thus, the proportions were close for AAN participants in the two studies, but our study did not include males, which makes a direct comparison of the two studies challenging. This study included hip structural analysis but not data from HRpQCT and µFEA.

In the study of adolescents and young adults with AN by Nagata et al., the authors also examined the impact of prior overweight history on bone outcomes and showed that a prior history of overweight was associated with greater BMD Z-scores [12]. The study of males with AN and AAN by Schorr et al. reported positive associations of BMI, appendicular lean mass, highest past BMI and lowest past BMI, and inverse association of duration of illness with BMD Z-scores at multiple sites [11]. In contrast, the study by Schorr et al. in women [28] reported that those with a history of overweight and obesity (but not those without this history) had lower tibial trabecular bone fraction, total and cortical vBMD and lower radial trabecular number than controls. This suggests that adult women with a history of overweight or obesity may fare worse than those without this history if they develop AAN. Further, AAN with amenorrhea (but not those who were eumenorrheic) had lower tibial and radial cortical thickness, total and cortical vBMD than controls. We did not consistently have information for previous overweight status in our participants, but we did examine associations with other measures.

For the full group, as expected, there were positive associations of BMI, percent median BMI, and lean mass with skeletal variables. This is consistent with skeletal loading from body weight and the pull of muscle on bone having bone anabolic effects [17,29,30]. Further, bone age, which provides an indication of pubertal status and the extent of exposure to sex steroids. was positively associated with radial and tibial cortical area and thickness, and cortical and total vBMD [31]. Estrogen exposure reduces endosteal bone resorption, which should result in greater cortical thickness and area, as observed in this study. As expected, later age of menarche (which indicates shorter duration of lifetime estrogen exposure) was associated with lower DXA measures of areal BMD Z-scores, and with lower cortical area and vBMD, lower trabecular thickness and vBMD, and lower total vBMD at both non-weight-bearing and weight-bearing sites [17,32]. This is consistent with endocrine effects of sex steroids on bone regardless of weight-bearing status [31]. A longer duration of illness was associated with negative effects at both the radius and tibia, consistent with expectations. However, contrary to expectations, duration of amenorrhea was not associated with most bone variables; this likely reflects enrollment of participants with irregular cycles in the study as the duration of amenorrhea in these women does not take into account varying length of cycles, which may also have a negative impact on bone health. The absence of an association of hours of exercise activity with skeletal measures likely reflects the fact that many of these young women were under instructions to refrain from exercise activity to optimize weight regain; thus, the reported measures of exercise activity did not necessarily reflect lifetime exposure to bone-loading activity.

Our study has several limitations. While areal BMD measures by DXA were available in a large number of participants, the subset that included HRpQCT and µFEA measures was smaller. However, our numbers are comparable for the latter to those in the study of adult women with AAN. Our data are also retrospective and cross-sectional, and a prospective study examining bone accrual rates in AAN vs. AN and controls could be very useful, particularly for an adolescent population. Notably, distinction between AN and AAN was based on operationalization of low weight by BMI and BMI percentile thresholds, which may be more flexibly applied in clinical practice. Thus, some participants may be diagnosed with AN and have a significantly low body weight within the context of their age, sex, and developmental trajectory but have a BMI ≥ 18.5 kg/m^2^ (adults) or ≥5th percentile (youth). Further, we did not consistently have reliable information for calcium and vitamin D intake and 25OHD levels, highest and lowest weight, and impact of weight loss in all participants, which needs to be addressed in future studies. Importantly, our study sample was very homogenous for race (predominantly White), and data are not generalizable across all races. However, our study has many strengths, including the fact that it is the first study reporting measures of vBMD, bone geometry, microarchitecture and strength estimates in adolescents with AAN compared with AN and controls. It is also the first study that compares girls with AAN to both AN and controls, thus providing a more comprehensive assessment of where these young women stand in relation to each other with regard to bone health.

## 5. Conclusions

Our study shows that adolescent and young adult women with AAN tend to have bone outcomes that are intermediate between AN and controls, and overall the distal tibia is impacted more than the distal radius. Importantly, a large proportion of AAN participants had low BMD Z-scores and run the risk for having suboptimal peak bone mass with an increased risk of osteoporosis and fractures in later life. While monitoring bone health is now routine practice for those with AN, this study highlights the importance of monitoring bone outcomes in AAN as well. Further, similar to AN [33], those with AAN should be actively managed with measures aimed at weight restoration and calcium and vitamin D supplementation, with consideration of estrogen replacement if they have amenorrhea with low BMD. Future studies that carefully study adolescents and young adult women with AAN are necessary to better understand the relative contributions of weight, history of overweight and underweight, menstrual status, and duration of illness on bone outcomes.

## Figures and Tables

**Figure 1 nutrients-15-03946-f001:**
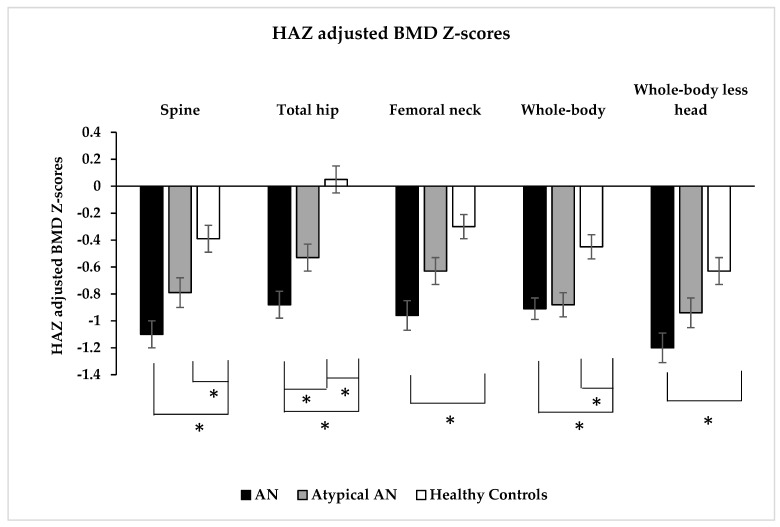
Height Z-score (HAZ) adjusted bone mineral density Z-scores at the spine, total hip, femoral neck, whole-body and whole-body less head (by dual-energy X-ray absorptiometry) in adolescent and young adult women with typical anorexia nervosa (AN), atypical AN (AAN) and healthy controls (HC). * *p* < 0.05.

**Figure 2 nutrients-15-03946-f002:**
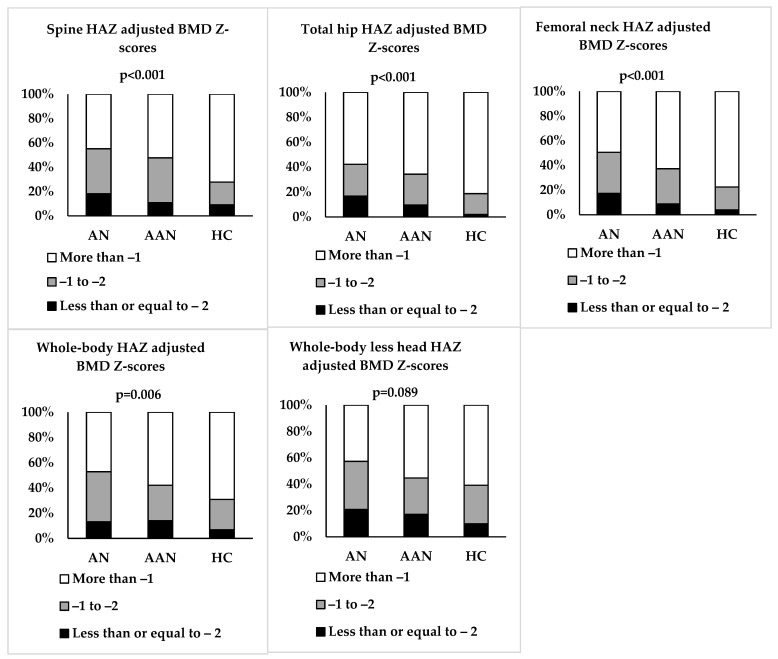
Proportion of adolescent and young adult women with typical anorexia nervosa (AN), atypical AN (AAN) and healthy controls (HC) whose height Z-score (HAZ) adjusted BMD scores are ≤−2, between −1 and −2, and >−1.

**Figure 3 nutrients-15-03946-f003:**
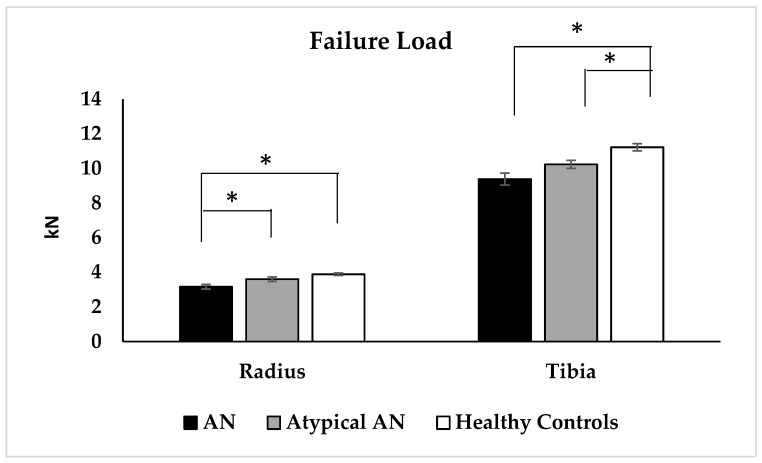
Failure load as assessed by microfinite element analysis at the distal radius and distal tibia in adolescent and young adult women with typical anorexia nervosa (AN), atypical AN (AAN) and healthy controls (HC). * *p* < 0.05.

**Table 1 nutrients-15-03946-t001:** Clinical characteristics and DXA measures of areal bone mineral density in the typical AN, atypical AN, and healthy control groups.

Characteristics	Typical AN = 141	Atypical AN N = 131	Healthy Controls (HC) N = 160	*p*-Value			
				Overall	Typical AN vs. Atypical AN	Atypical AN vs. HC	Typical AN vs. HC
**Age (years)**	17.5 ± 0.2 N = 141	17.0 ± 0.2 N = 130	17.2 ± 0.2 N = 160	0.202	0.220	0.715	0.606
**Bone age (years) in those 18 and younger**	16.5 ± 0.1 N = 130	16.3 ± 0.1 N = 123	16.5 ± 0.1 N = 159	0.433	0.483	0.574	0.977
**Race (White/Asian/Others) %**	87.4/3.9/8.7 N = 103	92.0/0/8.0 N = 113	70.3/13.3/16.4 N = 128	**<0.001**			
**Height (cm)**	164.3 ± 0.5 N = 140	164.1 ± 0.6 N = 130	163.7 ± 0.6 N = 159	0.709	0.958	0.865	0.693
**Weight (kg)**	44.8 ± 0.4 N = 141	50.9 ± 0.5 N = 131	57.3 ± 0.7 N = 159	**<0.001**	**<0.001**	**<0.001**	**<0.001**
**BMI (kg/m^2^)**	16.6 ± 0.1 N = 140	18.8 ± 0.1 N = 130	21.3 ± 0.2 N = 159	**<0.001**	**<0.001**	**<0.001**	**<0.001**
**Percent median BMI**	79.4 ± 0.4 N = 140	91.0 ± 0.4 N = 130	102.8 ± 0.9 N = 155	**<0.001**	**<0.001**	**<0.001**	**<0.001**
**Tanner stage breast**	5 (4, 5) N = 116	5 (4, 5) N = 113	5 (5, 5) N = 144	0.064	0.587	0.052	0.366
**Age at menarche (years)**	13.0 (12.0, 13.3) N = 113	12.1 (12.0, 13.0) N = 111	13.0 (12.0, 13.0) N = 128	0.340	0.351	0.952	0.495
**Proportion premenarchal (%)**	8.0% (10/125)	11.2% (14/125)	4.9% (7/143)	0.157	-	-	-
**Duration since diagnosis (months)**	12.0 (4.0, 30.0) N = 139	9.5 (3.0, 25.1) N = 126	-	0.315	-	-	-
**Duration of amenorrhea (months)**	2.0 (0.5, 8) N = 111	1.0 (0.4, 6.0) N = 98	-	**0.015**	-	-	-
**Proportion presenting with amenorrhea (%)**	46.0% (51/111)	33.0% (32/97)	-	0.057	-	-	-
**Exercise activity (hours per week)**	13.9 (8.1, 23.5) N = 102	14.9 (7.7, 22.1) N = 86	13.3 (7.8, 19.9) N = 96	0.717	0.933	0.858	0.726
**Lean mass (kg)**	34.9 ± 0.4 N = 84	37.8 ± 0.4 N = 81	41.0 ± 0.6 N = 117	**<0.001**	**<0.001**	**<0.001**	**<0.001**
**Fat mass (kg)**	8.9 ± 0.3 N = 84	13.1 ± 0.4 N = 81	17.4 ± 0.4 N = 117	**<0.001**	**<0.001**	**<0.001**	**<0.001**
**Percent body fat**	18.2 ± 0.5 N = 139	23.0 ± 0.4 N = 130	28.2 ± 0.4 N = 158	**<0.001**	**<0.001**	**<0.001**	**<0.001**
**DXA measures of areal BMD**							
**Spine BMD (g/cm^2^)**	0.88 ± 0.01 (n = 131)	0.91 ± 0.01 (n = 123)	0.96 ± 0.01 (n = 158)	**<0.001**	0.165	**0.002**	**<0.001**
**Total hip BMD (g/cm^2^)**	0.85 ± 0.01 (n = 109)	0.89 ± 0.01 (n = 121)	0.98 ± 0.01 (n = 127)	**<0.001**	**0.025**	**<0.001**	**<0.001**
**Femoral neck BMD (g/cm^2^)**	0.78 ± 0.01 (n = 102)	0.81 ± 0.01 (n = 114)	0.86 ± 0.01 (n = 127)	**<0.001**	0.086	**0.002**	**<0.001**
**Whole-body BMD (g/cm^2^)**	1.00 ± 0.01 (n = 140)	1.00 ± 0.01 (n = 130)	1.03 ± 0.01 (n = 158)	**<0.001**	0.981	**0.002**	**0.002**
**Whole-body less head BMD (g/cm^2^)**	0.87 ± 0.01 (n = 97)	0.88 ± 0.01 (n = 106)	0.92 ± 0.01 (n = 127)	**<0.001**	0.486	**<0.001**	**<0.001**

AN: typical anorexia nervosa; HC: healthy controls; BMI: body mass index; BMD: bone mineral density; significant *p*-values are indicated in bold font.

**Table 2 nutrients-15-03946-t002:** Clinical characteristics, HRpQCT and microfinite element analysis measures in the subset of typical AN, atypical AN (AAN) and healthy control (HC) groups that had HRpQCT assessment.

Characteristics	Typical AN (AN) N = 23	Atypical AN (AAN) N = 37	Healthy Controls (HC) N = 78	*p*-Value			
				Overall	AN vs. AAN	AAN vs. HC	AN vs. HC
Age (years)	19.3 ± 0.4	18.7 ± 0.3	19.1 ± 0.2	0.444	0.442	0.615	0.833
Bone age (years) in those 18 and younger	17.3 ± 0.3	17.3 ± 0.2	17.5 ± 0.1	0.489	0.999	0.573	0.646
Height (cm)	164.5 ± 1.5	163.7 ± 0.9	163.9 ± 0.8	0.911	0.904	0.985	0.938
Weight (kg)	46.1 ± 0.9	53.0 ± 0.7	59.3 ± 1.0	**<0.001**	**0.001**	**<0.001**	**<0.001**
Body mass index (kg/m^2^)	17.1 ± 0.2	19.8 ± 0.2	22.0 ± 0.3	**<0.001**	**<0.001**	**<0.001**	**<0.001**
Percent median BMI	79.5 ± 1.1	93.0 ± 0.81	102.9 ± 1.3	**<0.001**	**<0.001**	**<0.001**	**<0.001**
Tanner stage	4.6 ± 0.1	4.8 ± 0.1	4.9 ± 0.1	0.089	0.300	0.826	0.072
Age at menarche (years)	13.3 (12.0, 14.0)	13.0 (12.0, 13.3)	13.0 (12.0, 13.0)	0.116	0.308	0.773	0.108
Duration since diagnosis (months)	40.0 (15.0, 60.0)	13.5 (4.0, 29.8)	-	**0.005**	-	-	-
Duration of amenorrhea (months)	5.5 (0, 20.3)	1.5 (0, 6.3)	-	0.257	-	-	-
Exercise activity (hours/week)	13.8 (0.3, 23.5)	15.8 (7.2, 23.0)	12.0 (8.8, 16.6)	0.779	0.997	0.781	0.845
Lean mass (kg)	35.2 ± 0.7	37.7 ± 0.5	42.6 ± 0.7	**<0.001**	0.163	**<0.001**	**<0.001**
Fat mass (kg)	10.6 ± 0.6	14.5 ± 0.5	17.9 ± 0.5	**<0.001**	**0.001**	**<0.001**	**<0.001**
Percent body fat	22.2 ± 1.1	26.7 ± 0.7	29.2 ± 0.6	**<0.001**	**0.002**	**0.032**	**<0.001**
**Distal Radius**							
**Geometry**							
Cortical Area (mm^2^)	41.90 ± 1.71	46.30 ± 1.89	49.76 ± 1.28	**0.008 *^,†^**	0.285	0.257	**0.008 *^,†^**
% Cortical Area	16.63 (15.47, 19.26)	18.20 (14.98, 21.98)	19.65 (15.26, 23.61)	0.088	0.479	0.702	0.064
Trabecular Area (mm^2^)	207.4 (189.0, 241.4)	209.5 (180.1, 233.8)	208.3 (180.8, 242.3)	0.754	0.821	0.779	0.983
% Trabecular Area	83.11 ± 0.96	81.11 ± 0.97	80.46 ± 0.65	0.142	0.378	0.834	0.118
Cortical Thickness (mm)	0.64 (0.56, 0.71)	0.70 (0.58, 0.81)	0.78 (0.61, 0.88)	**0.033 ***	0.455	0.436	**0.027 ^†^**
**Microarchitecture**							
Cortical Porosity (%)	1.00 (0.80, 1.39)	0.84 (0.62, 01.29)	0.68 (0.45, 1.21)	**0.021**	0.704	0.118	**0.043**
Trabecular Thickness (mm)	0.06 (0.06, 0.07)	0.07 (0.06, 0.08)	0.07 (0.07, 0.08)	**0.013 ^†^**	0.066	0.759	**0.012 *^,†^**
Trabecular Separation (mm)	0.46 (0.43, 0.52)	0.44 (0.40, 0.50)	0.43 (0.39, 0.47)	**0.048 ***	0.480	0.394	**0.050**
Trabecular Number (1/mm)	1.92 (1.72, 2.05)	1.92 (1.76, 2.11)	1.99 (1.83, 2.18)	0.110	0.666	0.407	0.132
**Volumetric BMD**							
Cortical vBMD (mgHA/cm^3^)	801.0 (790.0, 841.9)	814.4 (783.9, 859.8)	840.8 (797.7, 870.2)	0.139	0.657	0.577	0.128
Trabecular vBMD (mgHA/cm^3^)	145.6 ± 6.7	162.0 ± 5.1	172.1 ± 3.9	**0.004 *^,†^**	0.156	0.278	**0.003 *^,†^**
Total vBMD (mgHA/cm^3^)	270.6 ± 9.0	298.6 ± 8.5	311.9 ± 6.2	**0.004 *^,†^**	0.111	0.407	**0.003 *^,†^**
**Strength estimates**							
Stiffness (kN/m)	58.8 (51.5, 69.7)	67.6 (60.4, 77.1)	74.5 (68.1, 84.6)	**<0.001 *^,†^**	0.069 ^†^	**0.028**	**<0.001 *^,†^**
Elastic Modulus	1574.0 (1430.0, 1870.0)	1839.5 (1579.8, 2048.8)	1937.0 (1648.9, 2147.4)	**0.002 *^,†^**	0.061 ^†^	**0.449**	**0.001 *^,†^**
Force of Fall (N)	2549.4 ± 15.4	2720.7 ± 15.4	2879.1 ± 20.9	**<0.001 ^†^**	**<0.001 ^†^**	**<0.001 ^†^**	**<0.001 ^†^**
Load-to-Strength Ratio (factor of risk)	861.8 (707.1, 969.3)	795.3 (703.1, 871.1)	749.6 (671.0, 846.5)	0.073	0.414	0.568	0.067
**Distal Tibia**							
**Geometry**							
Cortical Area (mm^2^)	96.63 ± 3.20	107.61 ± 2.74	121.67 ± 2.51	**<0.001 *^,†^**	0.096	**0.002 *^,†^**	**<0.001 *^,†^**
% Cortical Area	15.70 (12.96, 17.34)	17.57 (14.67, 19.28)	18.63 (15.31, 21.89)	**<0.001 *^,†^**	0.052	0.216	**0.001 *^,†^**
Trabecular Area (mm^2^)	536.7 ± 14.0	515.5 ± 13.3	532.5 ± 11.4	0.496	0.654	0.617	0.979
% Trabecular Area	84.64 ± 0.53	82.45 ± 0.60	80.96 ± 0.52	**<0.001 *^,†^**	0.110	0.165	**<0.001 *^,†^**
Cortical Thickness (mm)	0.99 ± 0.03	1.12 ± 0.03	1.23 ± 0.03	**<0.001 *^,†^**	0.076	**0.018 *^,†^**	**<0.001 *^,†^**
**Microarchitecture**							
Cortical Porosity (%)	2.34 (1.77, 3.05)	2.49 (1.92, 3.24)	1.47 (0.97, 2.39)	**<0.001 *^,†^**	0.974	**0.001 ^†^**	**0.016**
Trabecular Thickness (mm)	0.08 ± 0.00	0.09 ± 0.00	0.09 ± 0.00	0.551	0.752	0.682	0.994
Trabecular Separation (mm)	0.51 (0.46, 0.54)	0.46 (0.43, 0.51)	0.43 (0.39, 0.46)	**<0.001 *^,†^**	0.154	**0.012 ***	**<0.001 *^,†^**
Trabecular Number (1/mm)	1.69 (1.52, 1.90)	1.81 (1.68, 1.97)	1.95 (1.83, 2.12)	**<0.001 *^,†^**	0.254	**0.007 *^,†^**	**<0.001 *^,†^**
**Volumetric BMD**							
Cortical vBMD (mgHA/cm^3^)	851.4 (837.7, 887.5)	863.7 (844.1, 894.5)	878.6 (850.8, 909.7)	**0.031 *^,†^**	0.307	0.288	**0.046 ^†^**
Trabecular vBMD (mgHA/cm^3^)	172.4 ± 8.4	190.6 ± 4.6	198.9 ± 4.0	**0.006 *^,†^**	0.120	0.454	**0.004 *^,†^**
Total vBMD (mgHA/cm^3^)	279.4 ± 8.8	311.7 ± 5.9	329.5 ± 6.1	**<0.001 *^,†^**	**0.032 *^,†^**	0.148	**<0.001 *^,†^**
**Strength estimates**							
Stiffness (kN/m)	186.5 ± 7.5	204.8 ± 4.8	223.8 ± 4.4	**<0.001 *^,†^**	0.138	**0.024 *^,†^**	**<0.001 *^,†^**
Elastic Modulus	2308.0 ± 78.2	2564.9 ± 54.4	2710.9 ± 55.8	**<0.001 *^,†^**	0.070	0.217	**<0.001 *^,†^**

AN: typical anorexia nervosa; AAN: atypical AN; HC: healthy controls; vBMD: volumetric bone mineral density; significant *p*-values are indicated in bold font. * Indicates the overall *p*-value and *p*-values for comparisons of HRpQCT and micro-FEA measures across groups that remain significant after adjusting for race. ^†^ Indicates the overall *p*-value and *p*-values for comparisons of HRpQCT and micro-FEA measures across groups that remain significant after adjusting for height.

**Table 3 nutrients-15-03946-t003:** Associations between bone variables and possible determinants of these variables for the full group.

	BMI		Percent Median BMI		Lean Mass		Bone Age	
	r	*p*	r	*p*	r	*p*	r	*p*
**DXA variables**								
Lumbar spine BMD Z-score	0.307	**<0.001**	0.366	**<0.001**	0.082	0.232	0.011	0.832
Total hip BMD Z-score	0.460	**<0.001**	0.474	**<0.001**	0.330	**<0.001**	0.102	0.086
Femoral neck BMD Z-score	0.344	**<0.001**	0.378	**<0.001**	0.295	**<0.001**	0.080	0.179
Whole-body BMD Z-score	0.282	**<0.001**	0.302	**<0.001**	0.174	**0.009**	0.092	0.081
Whole-body less head BMD Z-score	0.350	**<0.001**	0.352	**<0.001**	0.231	**0.007**	0.157	**0.010**
**HRpQCT variables**								
**Radius**								
% Radial cortical area	0.191	**0.025**	0.151	0.083	−0.047	0.586	0.426	**<0.001**
Radial cortical thickness	0.242	**0.004**	0.202	**0.020**	0.068	0.427	0.430	**<0.001**
Radial trabecular thickness	0.250	**0.003**	0.263	**0.002**	0.198	**0.020**	0.146	0.089
Radial trabecular number	0.173	**0.043**	0.176	**0.043**	0.094	0.275	−0.081	0.346
Radial cortical vBMD	0.172	**0.044**	0.055	0.526	−0.012	0.887	0.653	**<0.001**
Radial trabecular vBMD	0.297	**<0.001**	0.311	**<0.001**	0.217	**0.011**	0.049	0.571
Radial total vBMD	0.301	**<0.001**	0.268	**0.002**	0.068	0.427	0.397	**<0.001**
Radial failure load	0.407	**<0.001**	0.440	**<0.001**	0.470	**<0.001**	0.114	0.189
**Tibia**								
% Tibial cortical area	0.366	**<0.001**	0.363	**<0.001**	0.042	0.625	0.232	**0.007**
Tibial cortical thickness	0.503	**<0.001**	0.502	**<0.001**	0.290	**<0.001**	0.244	**0.004**
Tibial trabecular thickness	0.019	0.824	0.024	0.787	0.019	0.825	0.140	0.103
Tibial trabecular number	0.377	**<0.001**	0.384	**<0.001**	0.383	**<0.001**	−0.094	0.276
Tibial cortical vBMD	0.268	**0.002**	0.173	**0.045**	−0.057	0.505	0.519	**<0.001**
Tibial trabecular vBMD	0.312	**<0.001**	0.323	**<0.001**	0.332	**<0.001**	0.051	0.551
Tibial total vBMD	0.410	**<0.001**	0.406	**<0.001**	0.189	**0.027**	0.220	**0.010**
Tibial failure load	0.515	**<0.001**	0.530	**<0.001**	0.671	**<0.001**	0.085	0.326
	**Age at menarche**		**Duration since diagnosis**		**Duration of amenorrhea**		**Exercise activity**	
	r	*p*	r	*p*	r	*p*	r	*p*
**DXA variables**								
Lumbar spine BMD Z-score	−0.365	**<0.001**	−0.107	0.114	−0.120	0.058	0.018	0.779
Total hip BMD Z-score	−0.212	**<0.001**	−0.073	0.317	−0.133	**0.044**	0.064	0.386
Femoral neck BMD Z-score	−0.253	**<0.001**	−0.039	0.601	−0.130	0.052	0.061	0.408
Whole-body BMD Z-score	−0.278	**<0.001**	−0.075	0.259	−0.073	0.242	0.070	0.264
Whole-body less head BMD Z-score	−0.239	**<0.001**	−0.017	0.827	−0.027	0.692	0.005	0.950
**HRpQCT variables**								
**Radius**								
% Radial cortical area	−0.190	**0.028**	−0.123	0.359	−0.063	0.536	−0.157	0.188
Radial cortical thickness	−0.146	0.093	−0.167	0.211	−0.041	0.686	−0.124	0.299
Radial trabecular thickness	−0.204	**0.018**	−0.337	**0.009**	−0.062	0.540	0.108	0.363
Radial trabecular number	−0.059	0.496	−0.065	0.627	−0.096	0.343	0.016	0.895
Radial cortical vBMD	−0.182	**0.035**	0.026	0.842	0.054	0.594	−0.033	0.784
Radial trabecular vBMD	−0.172	**0.046**	−0.289	**0.026**	−0.105	0.296	0.071	0.549
Radial total vBMD	−0.260	**0.002**	−0.225	0.087	−0.091	0.368	−0.073	0.542
Radial failure load	−0.135	0.120	−0.351	**0.007**	−0.116	0.254	0.022	0.857
**Tibia**								
% Tibial cortical area	−0.315	**<0.001**	−0.324	**0.012**	−0.112	0.268	0.071	0.553
Tibial cortical thickness	−0.250	**0.003**	−0.372	**0.004**	−0.133	0.188	0.129	0.279
Tibial trabecular thickness	−0.246	**0.004**	−0.186	0.160	0.056	0.580	0.047	0.692
Tibial trabecular number	−0.042	0.625	0.009	0.946	−0.115	0.254	0.105	0.376
Tibial cortical vBMD	−0.270	**0.002**	−0.141	0.286	0.033	0.746	0.106	0.370
Tibial trabecular vBMD	−0.244	**0.004**	−0.154	0.245	−0.071	0.482	0.063	0.596
Tibial total vBMD	−0.357	**<0.001**	−0.301	**0.020**	−0.093	0.357	0.081	0.497
Tibial failure load	−0.094	0.280	−0.277	**0.034**	−0.111	0.274	0.066	0.581

BMD: bone mineral density; vBMD: volumetric BMD; DXA: dual-energy X-ray absorptiometry; HRpQCT: high-resolution peripheral quantitative computed tomography. Significant *p*-values are indicated in bold font.

## Data Availability

The datasets used and analyzed during the current study are available from the principal investigators of the study on reasonable request.

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
