# Peer review of "Bone Density, Geometry, Structure and Strength Estimates in Adolescent and Young Adult Women with Atypical Anorexia Nervosa versus Typical Anorexia Nervosa and Normal-Weight Healthy Controls"

_nutrients, 2023, doi:10.3390/nu15183946_

Round 1

Reviewer 1 Report

This article has some important information about bone health in patients with AAN and AN in adolescent and young women. However, the authors should tell that more clearer picture of differ from AAN and AN. In introduction, the authors have described them, but the readers will be hard to understand.

Please edit how prone these patients will develop osteoporosis in middle age.

Author Response

We thank the Reviewer for their helpful comments regarding the manuscript. 

Comment 1: This article has some important information about bone health in patients with AAN and AN in adolescent and young women. However, the authors should tell that more clear picture of differ from AAN and AN. In introduction, the authors have described them, but the readers will be hard to understand.

Response to Comment 1: We thank the Reviewer for this suggestion. We have now expanded the first paragraph of the Discussion to indicate more clearly the differences between AAN and the other two groups. This paragraph now reads as follows (Lines 281-296):

“As hypothesized, we found that overall, AAN had bone outcomes that were better than AN but worse than HC, thus intermediate between AN and HC. This was true for total hip and femoral neck BMD DXA measures and for height adjusted BMD Z-scores at the spine, total hip and femoral neck. However, this was not consistent across measures. For example, for spine, whole body and whole body less head BMD, and for whole body and whole body less head height adjusted BMD Z-scores, measures did not differ between AN and AAN, but were lower in both AN and AAN compared to HC. For tibial parameters, AAN vs. HC had lower tibial cortical area, cortical thickness and trabecular number, greater trabecular separation and cortical porosity, and lower tibial stiffness and failure load (but did not differ from AN for these measures), while AAN had higher tibial total vBMD than AN.  For radial parameters, failure load was lower in AN vs. AAN, and radial stiffness was lower in AAN than HC. Similarly, many distal radius measures (by HRpQCT) did not differ between AN and AA (radial cortical area, thickness and porosity, trabecular thickness, trabecular and total vBMD, elastic modulus), but were adversely impacted in AN vs. HC. Other measures did not differ between AAN and HC, again suggesting that AAN were mostly intermediate between AN and HC for these measures.”

Comment 2: Please edit how prone these patients will develop osteoporosis in middle age.

Response to Comment 2: The Reviewer brings up an interesting point. While there are no predictive models for bone density in middle age based on such findings, we have included a sentence in the Conclusion about the potential of AAN to develop osteoporosis in later life (Lines 412-414.

Importantly, though a large proportion of AAN participants had low BMD Z-scores and run the risk for having suboptimal peak bone mass with an increased risk of osteoporosis and fractures in later life.”

Reviewer 2 Report

This is an interesting study that investigates the differences among AN, ANN, and HC with a sufficient sample size. On the other hand, it was difficult to understand what kind of specific development and contribution to treatment can be expected in the future, as I could not get out of the impression that it was just a list of measured facts.

Line238: For those who are not familiar with this field, this number of participants seems to be a sufficient number, and it is difficult to take it as not being a sufficient number of people. Reasons should be considered, such as the number of people to be surveyed and how many would seem sufficient.

Discussion: It is significant that they were able to measure various bone-related parameters in a large number of subjects in this study, and the differences from previous studies are clear. However, it was difficult to determine what specific therapeutic contributions could be made in the future based on the results of this study. A more detailed discussion of the significance of this discovery is required.

Author Response

We thank the Reviewer for the helpful comments.

Comment 1: This is an interesting study that investigates the differences among AN, ANN, and HC with a sufficient sample size. On the other hand, it was difficult to understand what kind of specific development and contribution to treatment can be expected in the future, as I could not get out of the impression that it was just a list of measured facts.

Response to Comment 1: The Reviewer brings up an important point- thank you. We have added a couple of sentences to the Conclusion to highlight the importance of these findings (Lines 414-418).  While monitoring bone health in now routine practice for those with AN, this study highlights the importance of monitoring bone outcomes in AAN as well. Further, similar to AN, those with AAN should be actively managed with lifestyle measures and calcium and vitamin D supplementation, with consideration of estrogen replacement if amenorrheic with low BMD.”

Comment 2: Line238: For those who are not familiar with this field, this number of participants seems to be a sufficient number, and it is difficult to take it as not being a sufficient number of people. Reasons should be considered, such as the number of people to be surveyed and how many would seem sufficient.

Response to Comment 2: Please note that only 23 AN and 37 AAN had HRpQCT assessments as opposed to the much larger numbers with DXA assessments. When each group was further divided in two subgroups based on those who did or did not present with amenorrhea, some subgroups had exceedingly small numbers.

Comment 3: Discussion: It is significant that they were able to measure various bone-related parameters in a large number of subjects in this study, and the differences from previous studies are clear. However, it was difficult to determine what specific therapeutic contributions could be made in the future based on the results of this study. A more detailed discussion of the significance of this discovery is required.

Response to Comment 3: Thank you for comment. As indicated in our response to Comment 1, we have added a couple of sentences to the Conclusion to highlight the importance of these findings (Lines 414-418).  While monitoring bone health in now routine practice for those with AN, this study highlights the importance of monitoring bone outcomes in AAN as well. Further, similar to AN, those with AAN should be actively managed with lifestyle measures and calcium and vitamin D supplementation, with consideration of estrogen replacement if amenorrheic with low BMD.”

Reviewer 3 Report

This is a good paper however I have a few concerns.

1) When did the recruitment take place?

2)I was challenged about how some of the subjects were identified and regrouped into the AAN category using the DSM-5 when they had previously been categorized by DSM- 4

3) I felt that the amount of data  presented,  while important may be overwhelming to the  reader caring for patients with AAN and AN

4) I could not find anywhere what DEXA scores are considered abnormal versus trending in the wrong direction

5) The conclusion should be more definitive- I finished reading the very long paper and did not take away much except- "I realize this is important but when and how should I get a DEXA scan in an AAN?" and ..."will insurance pay for this scan?"

Author Response

Comment 1: When did the recruitment take place?

Response to Comment 1: Thank you for asking the question. Recruitment occurred between 2000-2018 and this information is now included in the text (line 88).

Comment 2: I was challenged about how some of the subjects were identified and regrouped into the AAN category using the DSM-5 when they had previously been categorized by DSM- 4

Response to Comment 2: We thank the Reviewer for the opportunity to clarify this point. Regrouping was primarily based on weight and menstrual history. Our past studies screened a large number of adolescents and young adult women with restrictive eating disorders across a range of BMI and varying menstrual status. Because many participants had initially been characterized per DSM-IV criteria, all participants were recategorized per DSM-5 criteria for this analysis taking into account their body mass index (BMI) and menstrual history. Specifically, the new criteria for anorexia nervosa (AN) no longer require the presence of amenorrhea and include text guidance around operationalization of the low weight criterion based on BMI <18.5 kg/m2 for adults and 5th percentile for youth. Thus, participants were grouped as having AN if they had a BMI of <18.5 kg/m2 if ≥ 18 years old, or <5th percentile if <18 years old, along with other features of AN regardless of menstrual status. Participants were grouped as having atypical AN (AAN) if they had a BMI of ≥ 18.5 kg/m2 if ≥ 18 years old, or 5th percentile if <18 years old, along with other features of AN (and including a history of weight loss) but regardless of menstrual status.  We used an absolute BMI cutoff for adults and a BMI percentile cutoff for adolescents given established norms. This is now detailed in lines 88-100.

Comment 3: I felt that the amount of data presented,  while important may be overwhelming to the reader caring for patients with AAN and AN

Response to Comment 3: We respectfully submit that given that this is a research study, we have provided the basic details regarding bone endpoints in this paper. We expect the paper to be read by a variety of readers, including clinicians and researchers. For researchers that work in the area of bone metabolism, the information we are providing is the very least they would expect from a paper looking at bone outcomes. We hope that clinicians will benefit particularly from the Discussion and Conclusion sections that provides an interpretation of study findings and their implication.  We have thus elected to leave the information as is in the manuscript.

Comment 4: I could not find anywhere what DEXA scores are considered abnormal versus trending in the wrong direction

Response to Comment 4: Thank you for the question. BMD Z-scores ≤ -2 are considered low for all (1), and BMD Z-scores <-1 are considered low for those engaged in weight-bearing exercise (2). We have now added this to the Methods section (lines 132-134). This is also the rationale for looking at proportions of participants with BMD Z-scores <-2, between -1 and -2, and greater than -1.

Comment 5: The conclusion should be more definitive- I finished reading the very long paper and did not take away much except- "I realize this is important but when and how should I get a DEXA scan in an AAN?" and ..."will insurance pay for this scan?"

Response to Comment 5:  We thank the Reviewer for the comment, and the clinical application of the results is indeed that DXA scans should be obtained not just in patients with AN, but also in those with AAN. The ISCD guidelines recommend that in patients with primary bone disease, or at risk for secondary bone disease, a DXA should be performed when the patient may benefit from interventions to decrease their elevated risk of a clinically significant fracture, and the DXA results will influence that management (3). Patients with AN are known to have bone disease secondary to their underlying eating disorder and its body composition and hormonal associations, and identification of low bone density in these patients does influence their subsequent management- these patients thus qualify for insurance coverage for DXA assessments. We are now demonstrating that patients with AAN too are at risk for low bone density.  Identification of low bone density in this population should result in targeted psychological and behavioral interventions to address the underlying eating disorder and associated behaviors. Those with associated oligo-amenorrhea should likely also be considered for hormone replacement therapy. Papers such as ours should result in insurance coverage for DXA assessments for the indication of AAN (and not just for AN). This information is included in the Conclusions. We also call for additional research to determine the contributions of other factors that might impact bone outcomes in AAN (lines 426-433).

  1. Crabtree NJ, Arabi A, Bachrach LK, Fewtrell M, El-Hajj Fuleihan G, Kecskemethy HH, Jaworski M, Gordon CM. Dual-energy X-ray absorptiometry interpretation and reporting in children and adolescents: the revised 2013 ISCD Pediatric Official Positions. J Clin Densitom. 2014;17(2):225-242. http://www.ncbi.nlm.nih.gov/pubmed/24690232.
  2. De Souza MJ, Nattiv A, Joy E, Misra M, Williams NI, Mallinson RJ, Gibbs JC, Olmsted M, Goolsby M, Matheson G. 2014 Female Athlete Triad Coalition Consensus Statement on Treatment and Return to Play of the Female Athlete Triad: 1st International Conference held in San Francisco, California, May 2012 and 2nd International Conference held in Indianapolis, Indiana, May 2013. Br J Sports Med. 2014;48(4):289. http://www.ncbi.nlm.nih.gov/pubmed/24463911.
  3. Skeletal Health Assessment In Children from Infancy to Adolescence. https://iscdorg/learn/official-positions/pediatric-positions/#:~:text=In%20patients%20with%20primary%20bone,results%20will%20influence%20that%20management.

Reviewer 4 Report

Interesting paper.  This group, the AAN group, is a very unique and challenging group.  I look forward to seeing more.  Would be great to see some longitudinal data.  I have a couple of comments/questions.

1.  I would have liked to see a breakdown of who the subset of ppts were who had HR-pQCT..  mean ages, gender, ht, wt..  Could that be added to the results table? 

2.  With DXA you have clearly taken size into acct, thank you for that.  I see that you have used the Standard protocols for XCT1 for those scans.  Did you make any attempt to adjust for limb length? The geometrical measures may have a large size effect that has not been accounted for.  I know it is complicated to perform % offset scanning on that generation scanner...  I would suggest if you have forearm DXA on any of the XCT participants scanned that a forearm length adjustment could be made.  If you did measure tibia length, this would also be an idea to pursue.

But again, very interesting information overall.

Author Response

Overall Comment: Interesting paper.  This group, the AAN group, is a very unique and challenging group.  I look forward to seeing more.  Would be great to see some longitudinal data.  I have a couple of comments/questions.

Response to Overall Comment: We thank the Reviewer for the positive comments regarding the manuscript.

Comment 1:  I would have liked to see a breakdown of who the subset of ppts were who had HR-pQCT..  mean ages, gender, ht, wt..  Could that be added to the results table? 

Response to Comment 1: As requested by the Reviewer, we have now provided this information in the revised Table 2. Overall, the subset behaved very similarly to the full group for these and other variables. We have now included the DXA measures in Table 1 and renumbered the Tables accordingly.

Comment 2:  With DXA you have clearly taken size into acct, thank you for that.  I see that you have used the Standard protocols for XCT1 for those scans.  Did you make any attempt to adjust for limb length? The geometrical measures may have a large size effect that has not been accounted for.  I know it is complicated to perform % offset scanning on that generation scanner...  I would suggest if you have forearm DXA on any of the XCT participants scanned that a forearm length adjustment could be made.  If you did measure tibia length, this would also be an idea to pursue. But again, very interesting information overall.

Response to Comment 2: The Reviewer brings up an important point. Unfortunately, we do not have limb lengths for such adjustment. The groups did not differ for height, however, we now also present information in Table 2 regarding differences that remained significant after adjusting for height (indicated with the symbol). This is also now indicated in the text (lines 223-224 and 246; footnote of Table 2).

Round 2

Reviewer 2 Report

Modification was fine.

Author Response

We thank the Reviewer for their response.